# Semi-automatic Data Enhancement for Document-Level Relation Extraction with Distant Supervision from Large Language Models

**Junpeng Li**[*], **Zixia Jia**[*], **Zilong Zheng**[✉]
National Key Laboratory of General Artificial Intelligence, BIGAI
{lijunpeng,jiazixia,zlzheng}@bigai.ai
https://github.com/bigai-nlco/DocGNRE

## Abstract

Document-level Relation Extraction (DocRE), which aims to extract relations from a long context, is a critical challenge in achieving fine-grained structural comprehension and generating interpretable document representations. Inspired by recent advances in in-context learning capabilities emergent from large language models (LLMs), such as ChatGPT, we aim to design an automated annotation method for DocRE with minimum human effort. Unfortunately, vanilla in-context learning is infeasible for document-level Relation Extraction (RE) due to the plenty of predefined fine-grained relation types and the uncontrolled generations of LLMs. To tackle this issue, we propose a method integrating a Large Language Model (LLM) and a natural language inference (NLI) module to generate relation triples, thereby augmenting document-level relation datasets. We demonstrate the effectiveness of our approach by introducing an enhanced dataset known as DocGNRE, which excels in re-annotating numerous long-tail relation types. We are confident that our method holds the potential for broader applications in domain-specific relation type definitions and offers tangible benefits in advancing generalized language semantic comprehension.

## 1 Introduction

Document-level Relation Extraction (DocRE) is a task that focuses on extracting fine-grained relations between entity pairs within a lengthy context (Yao et al., 2019; Nan et al., 2020; Wang et al., 2020; Zhou et al., 2021; Zhang et al., 2021; Ma et al., 2023). The abundance of entity pairs in a document, coupled with a vast array of fine-grained relation types, makes DocRE inherently more challenging than sentence-level RE. The challenge is observed not only in model learning but also in human annotations.

---

[*]Equal contributions. Authors ordered alphabetically.
[✉] Correspondence to Zilong Zheng <zlzheng@bigai.ai>.

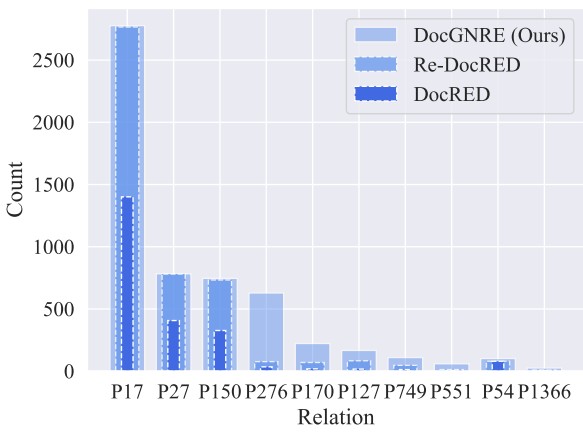

Figure 1: Counts of relation types in different datasets.

The original document-level RE dataset DocRED (Yao et al., 2019) has been recognized for its false negative issue and subsequently re-annotated to address this concern by supplementing a significant number of relation triples. Notably, two representative works, Huang et al. (2022) and Tan et al. (2022b), have contributed to this re-annotation process. Huang et al. (2022) undertook manual annotation from scratch, employing two expert annotators to annotate 96 documents. On the other hand, Tan et al. (2022b) utilized pre-trained RE models in conjunction with manual revision to construct Re-DocRED. Despite their contributions to supplementing relations for DocRED, both methods have certain limitations. **First**, achieving complete manual annotation is challenging: each document within this dataset contains an average of 19.5 entities, requiring consideration of approximately 37,000 candidate triples (including 97 relation types, including NULL). **Second**, the supplementary annotations are derived from the existing data distribution: Tan et al. (2022b) first pre-trained a RE model with distantly supervised data from DocRED, then utilized this model to predict triple candidates. Such a process may introduce model bias, potentially resulting in the exclusion of sparse relations that exist beyond the scope of the existing

data distribution. Figure 1 illustrates the counts of some relation types across various datasets. It is evident that the supplementary annotations in Re-DocRED exhibit a distribution similar to that of the DocRED test set.

In light of the limitations inherent in prior annotation techniques and the imperative to enhance the completeness of DocRE datasets, we propose a novel approach aimed at augmenting the Re-DocRED dataset through the utilization of the powerful generalization capabilities of Large Language Models (LLMs). By leveraging our method, as shown in Figure 1, our revised test set, named DocGNRE, exhibits the advantage of re-annotating a greater number of long-tail types, such as P276 and P551.

Recent studies have utilized GPT (Floridi and Chiriatti, 2020; Chan, 2023) for various structural prediction tasks, such as named entity prediction and relation extraction (Dunn et al., 2022; Gutiérrez et al., 2022; Wang et al., 2023; Liu et al., 2023; Xu et al., 2023), as well as text classification labeling (Gilardi et al., 2023; Törnberg, 2023). Notably, researchers such as Wan et al. (2023); Wadhwa et al. (2023) have demonstrated the effectiveness of in-context learning by incorporating prompts containing suitable example demonstrations for RE tasks. However, it is worth noting that without explicit instructions, GPT may generate uncontrolled relations that do not align with predefined types. Therefore, recent methods only work in sentence-level RE and especially highlight one distinguished challenge for LLM in-context learning: unable to fitting detailed instructions for long-context documents (Wadhwa et al., 2023).

To align GPT-generated relations and predefined relation types, we first combine Natural Language Inference (NLI) models (MacCartney, 2009) with GPT to solve zero-shot DocRE. The results show that although GPT generations only hit partial ground truth of Re-DocRED, it detects substantial external valid relation triples (details in Sec. 3.1). Therefore, we design a pipeline framework to further complement the test set of Re-DocRED and automatically generate distant training annotations by combining GPT and NLI modules. To verify that we supplement many relation triples beyond the scope of the original data distribution, we test previous models in our DocGNRE test set. Additionally, we train the state-of-the-art (SOTA) model using our distant training dataset.

Our contributions can be summarized as follows:
▷ We conduct a quantitative analysis to evaluate the performance of GPT in zero-shot document-level RE.
▷ We propose a novel framework that integrates an NLI module and an LLM to automatically generate distant relation triples, supplementing existing DocRE datasets.
▷ We create an enhanced version of the Re-DocRED test set, named DocGNRE, with minimal human intervention, ensuring high quality. Additionally, we augment the Re-DocRED training set by supplementing it with distant relation triples automatically generated by our framework[1], referring to Tabel 1.

## 2   LLM Enhanced Automatic Data Generation

Our approach consists of two main procedures: constructing prompts for LLM to generate relations triples as proposals and employing Natural Language Inference (NLI) models to align generated relations with predefined relation types. Fig. 2 shows the whole framework. In the first procedure, we observed that even though we imposed restrictions on the entity and relation lists in the prompts, LLMs (both GPT-3.5 and GPT-4) still generated triples that fell outside of our intended constraints. Furthermore, we found that the generated relations expressed by contextual words in the document were more accurate than those with the restricted relation types. Based on these insights and to fully leverage the potential of LLMs, hereby generating more accurate relation triples, we removed the restrictions of specific relation types for LLMs. Instead, we subsequently utilize an NLI module to map the generated relations to the predefined relation types in the Re-DocRED dataset.

### 2.1   GPT Results as Proposals

We select GPT-3.5 (gpt-3.5-turbo) as our LLM module, considering a balance between cost and performance. Given that the original DocRED dataset provides an entity list for each document, we constrain the responses of GPT to utilize only the entities present in the provided list.

**Prompt Construction**   As shown in Figure 2, the prompt consists of a generation demonstration and a specific context followed by a corresponding

---

[1]Our dataset is publicly available at `https://github.com/bigai-nlco/DocGNRE`.

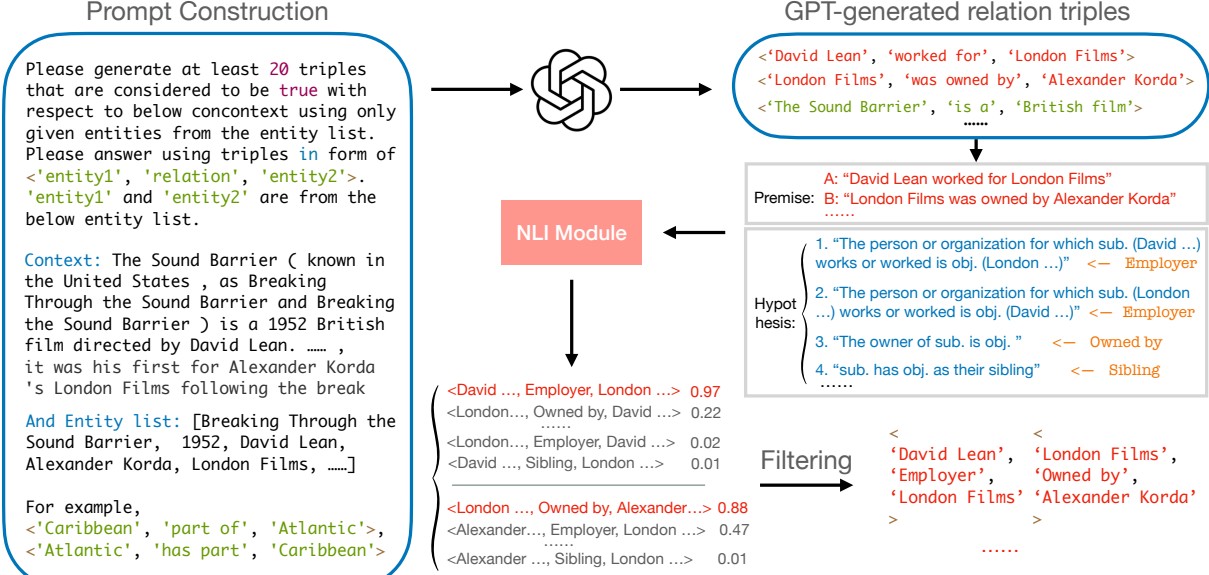

Figure 2: The automatic data generation framework and an exemplar document. The green triple in GPT-generated relation triples will be filtered because its object entity is out of the given entity list.

entity list. We notice that as the generated content by LLMs became longer, the accuracy decreased. To mitigate this, we set "at least 20 triples" in the initial prompt [2]. To generate more additional triples, we employ an iterative approach by feeding the previous GPT answers as input while instructing GPT to "Please keep generating 20 more triples using only the given entities from the entity list". However, despite providing the entity list in the prompt, we observed that undesired triples with incorrect entity pairs still occurred. To address this, we implemented a filtering process to remove these undesired triples. Consequently, all the remaining triples are treated as proposals and later aligned using the NLI module.

## 2.2 NLI Module as an Annotator

In this procedure, our goal is to map the relations generated by GPT to predefined types. To achieve this, a reasonable approach is to align the semantic meaning of relations. Therefore, we employ a NLI model, which has demonstrated effectiveness in assessing factual consistency (Honovich et al., 2022). The NLI model takes two sentences as input, typically referred to as the premise and the hypothesis. It assigns a score to each term, indi-

cating whether it signifies entailment, neutrality, or contradiction. If the term "entailment" receives the highest score, the model concludes that the two sentences are factually consistent.

**Premise and Hypothesis Construction** In our framework, we take each GPT-generated triple as the premise and replace the relation in such triple with a specific predefined relation type as the hypothesis. Remember that our purpose is to map each GPT-generated relation proposal to a predefined relation type. Hence, we should enumerate the hypothesis constructed by each specific type in the predefined set to calculate the entailment scores with the corresponding premise and choose the **ONE** with the highest score. Moreover, we observe that the GPT-generated relation may correspond to an inverse predefined relation type. For example, if the predefined relation set contains the "employee" type rather than "employer", the GPT-generated triple "<David Lean, worked for, London Films>" will correspond to "<London Films, employee, David Lean>" rather than "<David Lean, employee, London Films>". Therefore, for each generated relation proposal as a premise, we construct $96*2 = 192$ possible hypotheses, where 96 is the size of the predefined relation set without NULL, and double means we change the subject and object entities for each predefined type. Specifically, take a triple $< e_1, r_{gpt}, e_2 >$ generated from GPT as an example premise, given the predefined relation set $\{r_1, r_2, ...\}$, we construct candidate hypotheses $\{< e_1, r_1, e_2 >, < e_2, r_1, e_1 >, < e_1, r_2, e_2 >, <$

---

[2]We choose the "20" number because it is a trade-off between ensuring accuracy and the quantity of generated relations. We find that the first 20 or so relations generated by GPT-3.5 (gpt-3.5-turbo) exhibit a relatively promising level of quality. The "at least 20" could be replaced by "no more than" and "a maximum of". We discovered that GPT-3.5 generates comparable numbers and quality of triples using all of these expressions.

|       |                  | # Doc | # Ent  | # Tri   |
|-------|------------------|-------|--------|---------|
|       | Re-DocRED        | 500   | 9,779  | 17,448  |
| Test  | DocGNRE (Ours)   | 500   | 9,779  | 19,526  |
|       | Δ                | 0     | 0      | 2,078   |
|       | Re-DocRED        | 3,053 | 59,359 | 85,932  |
| Train | Re-DocRED+GPT    | 3,053 | 59,359 | 96,505  |
|       | Re-DocRED+more GPT | 3,053 | 59,359 | 103,561 |

Table 1: Comparison of relation statistics between Re-Docred and DocGNRE. The test set has been verified by human annotators. The GPT-generated triples (+GPT) on the train set are distant.

$e_2, r_2, e_1 >, ...\}$.

Because the NLI model is pretrained with natural language sentences, we convert the triples to natural sentences.

▷  Most of the GPT-generated relations are themselves in natural language, so each triple's subject entity, relation, and object entity are directly concatenated to get a natural sentence.

▷  The predefined relation types typically are abstractive. To make the hypothesis precisely convey the meaning of each relation type, we integrate the description of each relation type with subject and object entities. Hypothesis for each relation type can be found in Appendix C.

**Entail Scores from NLI Model**  We use the T5-based NLI model[3] in this paper for its powerful generalizability. T5-XXL (Raffel et al., 2020) is a generative model, which identifies "Entailment" and "No entailment" by generating two sequences in the inference stage. We leverage the probabilities of such two sequences omitting the start and end tokens to calculate the entailment scores used for sorting predefined relation types. Details can be found in Appendix A.

**Post Processing**  To ensure the high quality of newly produced relation triples, we ultimately retain those hypothesis triples that should satisfy all the following principles:

▷ The entity types of subject and object entities satisfy the type constraints of the relation.
▷ Get the highest entailment scores.
▷ Get the entailment scores of more than 0.6.

Note that some of the GPT-generated relations may be exactly those in the predefined set of relation types. We do not need to map these generated triples via our NLI module and just add them into the final selected triples set.

---

[3] https://huggingface.co/google/t5_xxl_true_nli_mixture

Through above procedures, we process each document of the Re-DocRED train set to produce additional distant relation triples. For the Re-DocRED test set, after acquiring distant relation triples, each distant triple will be conducted through human verification. Two annotators are asked to answer whether the relation triples can be inferred according to the provided documents. A third annotator will resolve the conflicting annotations. Specifically, we use Mechanical Turk for human annotations. In order to ensure that annotators possessed a significant level of qualification, prospective annotators were required to meet the following criteria:
• "HIT Approval Rate(%) for all Requesters' HITs" > 95.
• "Number of HITs Approved" > 1000.
• "Location" is one of {United States, Canada, Great Britain, Australia, Singapore, Ireland, New Zealand}.

The first two indicators are calculated by Mechanical Turk according to one's historical performance and the last one aims to promise English proficiency of the annotators. Finally, the acceptance rate of the NLI-selected relations in the test set is 71.3%. We provide a more accurate and complete test set with the addition of 2078 triples than Re-DocRED. Detailed statistics of our datasets can be found in Table 1.

## 3  Experiments

### 3.1  Zero-shot Document-level RE

Our framework can obviously be used to predict document-level relations directly. Therefore, in the first experiment, we explore the GPT performance on the zero-shot document-level RE. Table 2 shows the results. As far as we know, we are the first to report these results on document-level RE.

We have three observations based on Table 2: i) **Pure GPT-3.5 (without our NLI module) only hits rare ground truth.** As aforementioned, GPT generates most relations expressed by natural language, which do not exactly match with the ground truth, even though some of these relations represent the same meaning as ground truth. So the exact-match F1 scores are unsatisfactory; ii) **NLI module can improve pure GPT performance.** With NLI module mapping GPT answers to predefined types, GPT-3.5 (gpt-3.5-turbo) predicts a small portion of ground truth triples that are manually annotated (5.77 recall in the Re-DocRED test set). The reason may be that we ask it to generate multiple relations

| Method | DocRED | | | Re-DocRED | | | DocGNRE | | |
|---|---|---|---|---|---|---|---|---|---|
| | P | R | F1 | P | R | F1 | P | R | F1 |
| GPT-3.5 Only | 7.34 | 4.53 | 5.6 | 13.12 | 2.85 | 4.68 | 13.97 | 2.71 | 4.54 |
| GPT-3.5 + NLI (w/o. rel des) | 13.9 | 10.29 | 11.82 | 23.57 | 6.14 | 9.74 | 42.91 | 9.9 | 16.2 |
| GPT-3.5 + NLI (w. rel des) | 14.61 | 9.8 | 11.73 | 24.45 | 5.77 | 9.33 | 72.71 | 15.32 | 25.31 |

Table 2: Results of zero-shot document-level RE. We test the three test sets: DocRED, Re-DocRED, and our DocGNRE. "rel des" means relation description. "P" and "R" refer to precision and recall respectively.

| Train set | DocGNRE (D) | | | Re | D - Re |
|---|---|---|---|---|---|
| | P | R | F1 | R | R |
| DocRED † | **90.3** | 27.83 | 42.55 | 31.04 | 0.91 |
| DocRED ‡ | 84.52 | 32.1 | 46.52 | 35.73 | 1.54 |
| Re-DocRED † | 81.45 | 56.98 | 67.05 | 63.59 | 1.40 |
| Re-DocRED ‡ | 85.0 | 64.29 | 73.21 | 71.69 | 2.17 |
| Doc + GPT † | 84.54 | 27.9 | 41.96 | 29.72 | 12.66 |
| Doc+ GPT ‡ | 84.07 | 34.86 | 49.2 | 37.22 | 15.08 |
| Doc + mGPT † | 80.65 | 28.89 | 42.54 | 30.02 | 19.39 |
| Doc + mGPT ‡ | 79.29 | 36.31 | 49.75 | 38.09 | 21.4 |
| Re+ GPT † | 83.66 | 57.62 | 68.24 | 62.87 | 13.53 |
| Re+ GPT ‡ | 84.92 | 63.86 | 72.9 | 70.0 | 12.29 |
| Re+ mGPT † | 81.71 | 58.23 | 68.0 | 62.74 | 20.36 |
| Re+ mGPT ‡ | 80.93 | **66.98** | **73.29** | **72.36** | **21.86** |

Table 3: Results of DREEAM model (Ma et al., 2023) training with different settings. All results are averaged by three runs. "mGPT" means "more GPT" that we carry out two iterative processes as mentioned in Sec. 2.1. "Doc" and "Re" are abbreviation of "DocRED" and "Re-DocRED". "D-Re" refers to our supplementary triple set (the remaining triples obtained by removing triples of Re-DocRED from DocGNRE. † means BERT-base (Devlin et al., 2018) and ‡ means RoBERTa-large (Liu et al., 2019).

at once by one prompt rather than enumerate all entity pairs to ask for relations one by one (which is too costly and time-consuming to execute for plenty of entities on document-level RE). But from human verification, the accuracy of NLI-selected triples has been proven relatively high (72.71 precision in our supplementary test set DocGNRE), which illustrates that most triples predicted by our framework are the supplementary of Re-DocRED; iii) **Relation descriptions can guide NLI to output expected relations.** With relation descriptions to construct hypothesises, the performance is further improved (25.31 *vs*. 16.2) in our DocGNRE.

### 3.2 Training with Distant Triples

We test the SOTA document-level RE model (Ma et al., 2023) on our DocGNRE and retrain it with our distant training set. All experiment settings are the same as Ma et al. (2023) except for training data in the +GPT setting. Table 3 shows the results. We can find that **i) the recall of previous models on our DocGNRE drops**, which demonstrates the

difficult prediction on our supplementary test relation triples when the model is only trained with the training set of DocRED or ReDocRED; **ii) The recall scores on all the test sets are improved with directly supervised training on our training set** (which exhibits the capability to predict additional ground truth instances), even though our distantly supervised data is somewhat noisy. Designing more advanced methods to leverage our distant training set is taken in future work.

In addition, we conducted experiments using two other DocRE models, ATLOP (Zhou et al., 2021) and KD-DocRE (Tan et al., 2022a), by leveraging their officially provided code. Experimental results of ATLOP and KD-DocRE show a similar tendency to DREEAM. Detailed results are in Appendix B.

## 4 Conclusion

LLMs face challenges in extracting fine-grained relations within lengthy contexts. To address this limitation, we present a novel framework that integrates an NLI module in this work. With our framework, we improve the performance of GPT in zero-shot document-level RE. Above all, our framework enhances the automatic data generation capability with minimum human effort. We supplement the existing DocRE dataset, providing a complete test set DocGNRE and a distant training set. Given the inherent presence of false negative instances in numerous RE datasets, particularly those constructed through a recommend-revise scheme or distant supervision, we believe our framework possesses a broad utility that extends to a wider array of datasets.

## Limitations

The limited generated length of LLMs causes the limitation of our methods. There is a specific upper limit on the number of relation triples that can be generated for each document. Therefore, our framework is an excellent data supplement method rather than a perfect zero-shot predictor.

## Acknowledgements

This work is supported in part by the National Key R&D Program of China (2021ZD0150200) and the National Natural Science Foundation of China (62376031).

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

| Train set | DocGNRE (D) | | | Re | D - Re |
|---|---|---|---|---|---|
| | P | R | F1 | R | R |
| DocRED † | 87.9 | 28.6 | 43.2 | 31.9 | 0.9 |
| DocRED ‡ | **89.8** | 31.1 | 46.2 | 34.6 | 1.5 |
| Doc + GPT † | 82.9 | 29.3 | 43.2 | 31.2 | 13.0 |
| Doc + GPT ‡ | 84.4 | 32.2 | 46.6 | 34.4 | 13.3 |
| Doc + mGPT † | 79.6 | 30.2 | 43.8 | 31.5 | 19.54 |
| Doc + mGPT ‡ | 79.3 | **34.2** | **47.7** | **35.7** | **21.1** |

Table 4: Results of ATLOP model (Zhou et al., 2021) training with different settings. † means BERT-base (Devlin et al., 2018) and ‡ means RoBERTa-large (Liu et al., 2019).

| Train set | DocGNRE (D) | | | Re | D - Re |
|---|---|---|---|---|---|
| | P | R | F1 | R | R |
| DocRED † | **82.4** | 32.9 | 47.0 | 36.6 | 1.6 |
| DocRED ‡ | 76.9 | 37.8 | 50.3 | 41.9 | 2.7 |
| Doc + GPT † | 74.3 | 36.1 | 48.5 | 38.3 | 17.9 |
| Doc + GPT ‡ | 77.1 | 37.4 | 50.4 | 39.7 | 18.8 |
| Doc + mGPT † | 65.3 | 39.2 | 48.9 | 40.0 | **31.9** |
| Doc + mGPT ‡ | 68.8 | **41.2** | **51.6** | **42.5** | 30.8 |

Table 5: Results of KD-DocRE model (Tan et al., 2022a) training with different settings. † means BERT-base (Devlin et al., 2018) and ‡ means RoBERTa-large (Liu et al., 2019).

## A  NLI Score

We choose the T5-based NLI model released by Google in this paper. As mentioned earlier, T5-XXL is a generative model. The model identifies "Entailment" and "No entailment" by generating two sequences in the inference stage. During inference, the sequence "<pad>_0" identifies "No entailment", and the sequence "<pad>1<pad>" identifies "Entailment". Because both the first tokens are "<pad>", and when the first three tokens have been determined, the prediction of the last token has a high probability, so we do not consider the first and last token when calculating the NLI score. To obtain the NLI score, we obtain the logits of four subsequences ("_0", "_", "10", "1") and perform a softmax operation to obtain the corresponding probabilities. The score of "_0" sequence corresponds to the score of "No entailment". The score of "1" sequence corresponds to the score of "Entailment". To further distinguish the NLI score among the constructed triples, we subtract the score of "No entailment" from the score of "Entailment" to fuse the two scores as the final scores.

## B  Model Results

Experimental results of ATLOP and KD-DocRE are shown in Table 4 and Table 5 respectively.

## C  Hypothesis Construction of Relation

We provide hypothesis construction of predefined relations in DocRED and Re-DocRED with Wikidata Id and Name in Table 6.

| Wikidata ID | Name | Hypothesis Construction |
|---|---|---|
| P6 | head of government | The head of the executive power of the governmental body sub. is obj. |
| P17 | country | The sovereign state of this item sub. is obj. |
| P19 | place of birth | The birth location of the person, animal or fictional character sub. is obj. |
| P20 | place of death | The death location of the person, animal or fictional character sub. is obj. |
| P22 | father | The father of sub. is obj. |
| P25 | mother | The mother of sub. is obj. |
| P26 | spouse | The spouse of sub. is obj. |
| P27 | country of citizenship | obj. is a country that recognizes sub. as its citizen |
| P30 | continent | obj. is the continent of which sub. is a part |
| P31 | instance of | obj. is that class of which sub. is a particular example and member. (sub. typically an individual member with proper name label) |
| P35 | head of state | obj. is the official with the highest formal authority in the country/state sub. |
| P36 | capital | obj. is the primary city of the country/state sub. |
| P37 | official language | obj. is the language designated as official by sub. |
| P39 | position held | sub. currently or formerly holds the position or public office obj. |
| P40 | child | sub. has obj. as their offspring son or daughter |
| P50 | author | The main creator(s) of the written work sub. is(are) obj. |
| P54 | member of sports team | The sports team or club that sub. represents or formerly represented is obj. |
| P57 | director | The director of this film, TV-series, stageplay or video game is obj. |
| P58 | screenwriter | The author(s) of the screenplay or script for this work sub. is(are) obj. |
| P69 | educated at | The educational institution attended by sub. is obj. |
| P86 | composer | The person(s) who wrote the music sub. is(are) obj. |
| P102 | member of political party | The political party of which this politician sub. is or has been a member is obj. |
| P108 | employer | The person or organization for which sub. works or worked is obj. |
| P112 | founded by | The founder or co-founder of this organization, religion or place sub. is obj. |
| P118 | league | The league in which the team or player sub. plays or has played in is obj. |
| P123 | publisher | The organization or person responsible for publishing books, periodicals, games or software sub. is obj. |
| P127 | owned by | The owner of sub. is obj. |
| P131 | located in the administrative territorial entity | sub. is located on the territory of the following administrative entity obj. |
| P136 | genre | The creative work sub.'s genre is obj. |
| P137 | operator | The person or organization that operates the equipment, facility, or service sub. is obj. |
| P140 | religion | The religion of a person, organization or religious building, or associated with sub. is obj. |
| P150 | contains administrative territorial entity | The direct subdivisions of an administrative territorial entity sub. has obj. |

| Wikidata ID | Name | Hypothesis Construction |
|---|---|---|
| P155 | follows | The immediately prior item in some series of which sub. is part is obj. |
| P156 | followed by | The immediately following item in some series of which sub. is part is obj. |
| P159 | headquarters location | The specific location where sub.'s headquarters is or has been situated is obj. |
| P161 | cast member | The actor performing live sub. for a camera or audience has obj. |
| P162 | producer | The producer(s) of this film or music work sub. is(are) obj. |
| P166 | award received | The award or recognition received by a person, organization or creative work sub. is obj. |
| P170 | creator | The maker of a creative work sub. is obj. |
| P171 | parent taxon | The closest parent taxon of the taxon sub. is obj. |
| P172 | ethnic group | sub.'s ethnicity is obj. |
| P175 | performer | The performer involved in the performance or the recoding of the work sub. is obj. |
| P176 | manufacturer | The manufacturer or producer of the product sub. is obj. |
| P178 | developer | The organization or person that developed sub. is obj. |
| P179 | series | The series which contains sub. is obj. |
| P190 | sister city | sub. and obj. are twin towns, sister cities, twinned municipalities |
| P194 | legislative body | The legislative body governing sub. is obj. |
| P205 | basin country | The country that have drainage to/from or border the body of water sub. has obj. |
| P206 | located in or next to body of water | sub. is located in or next to body of water obj. |
| P241 | military branch | The branch to which the military unit, award, office, or person sub. belongs is obj. |
| P264 | record label | The brand and trademark associated with the marketing of subject music recordings and music videos sub. is obj. |
| P272 | production company | The company that produced this film, audio or performing arts work sub. is obj. |
| P276 | location | The location of the item, physical object or event sub. is within is obj. |
| P279 | subclass of | All instances of sub. are instances of obj. |
| P355 | subsidiary | The subsidiary of a company or organization sub. has obj. |
| P361 | part of | obj. has part or parts sub. |
| P364 | original language of work | The language in which the film or a performance work sub. was originally created is obj. |
| P400 | platform | The platform for which the work sub. has been developed or released / specific platform version fo the software sub. developed is obj. |
| P403 | mouth of the watercourse | The body of water to which the watercourse sub. drains is obj. |
| P449 | original network | The network(s) the radio or television show sub. was originally aired on has obj. |
| P463 | member of | The organization or club to which sub. belongs is obj. |
| P488 | chairperson | The presiding member of the organization, group or body sub. is obj. |
| P495 | country of origin | The country of origin of the creative work sub. is obj. |
| P527 | has part | sub. has part or parts obj. |
| P551 | residence | The place where the person sub. is, or has been, resident is obj. |
| P569 | date of birth | The date on which sub. was born is obj. |
| P570 | date of death | The date on which sub. died is obj. |

| Wikidata ID | Name | Hypothesis Construction |
|---|---|---|
| P571 | inception | The date or point in time when the organization/subject sub. was founded/created is obj. |
| P576 | dissolved, abolished or demolished | The date or point in time on which the organization sub. was dissolved/disappeared or the building sub. demolished is obj. |
| P577 | publication date | The data or point in time the work sub. is first published or released is obj. |
| P580 | start time | The time the item sub. begins to exist or the statement sub. starts being valid is obj. |
| P582 | end time | The time the item sub. ceases to exist or the statement sub. stops being valid is obj. |
| P585 | point in time | The time and date sub. took place, existed or the statement sub. was true is obj. |
| P607 | conflict | The battles, wars or other military engagements in which the person or item sub. participated is obj. |
| P674 | characters | The characters which appear in sub. has obj. |
| P676 | lyrics by | The author of song lyrics sub. is obj. |
| P706 | located on terrain feature | sub. is located on the specified landform obj. |
| P710 | participant | The person, group of people or organization that actively takes/took part in the event sub. has obj. |
| P737 | influenced by | The person, idea sub. is informed by obj. |
| P740 | location of formation | The location where the group or organization sub. was formed is obj. |
| P749 | parent organization | The parent organization of the organization sub. is obj. |
| P800 | notable work | The notable scientific, artistic or literary work, or other work of significance among sub.'s works is obj. |
| P807 | separated from | sub. was founded or started by separating from identified object obj. |
| P840 | narrative location | The narrative of the work sub. is set in the location obj. |
| P937 | work location | The location where persons or organization sub. were actively participating in employment, business or other work is obj. |
| P1001 | applies to jurisdiction | The institution, law or public office sub. belongs to or has power over or applies to the country, state or municipality obj. |
| P1056 | product or material produced | The material or product produced by the government agency, business, industry, facility, or process sub. is obj. |
| P1198 | unemployment rate | The portion of the workforce population that is not employed of sub. is obj. |
| P1336 | territory claimed by | The administrative divisions that claim control of the given area sub. is obj. |
| P1344 | participant of | The event that the person or the organization sub. was a participant in is obj. |
| P1365 | replaces | The person or item sub. replaces obj. |
| P1366 | replaced by | The person or item obj. replaces sub. |
| P1376 | capital of | sub. is capital of obj. |
| P1412 | languages spoken, written or signed | The language(s) that the person sub. speaks or writes is obj. |
| P1441 | present in work | The work in which the fictional entity or historical person sub. is present is obj. |
| P3373 | sibling | sub. has obj. as their sibling |

Table 6: Relation list, including Wikidata IDs, Names and Hypothesis Construction of relations